# Detection of Abnormal SIP Signaling Patterns: A Deep Learning Comparison

**Diogo Pereira** [1,2,*,†] **and Rodolfo Oliveira** [1,2,†]

1 Departamento de Engenharia Electrotécnica e de Computadores, Faculdade de Ciências e Tecnologia, FCT/UNL, Universidade Nova de Lisboa, 2829-516 Caparica, Portugal; rado@fct.unl.pt
2 Instituto de Telecomunicacoes, 1049-001 Lisbon, Portugal
* Correspondence: dfca.pereira@campus.fct.unl.pt
† These authors contributed equally to this work.

**Abstract:** This paper investigates the detection of abnormal sequences of signaling packets purposely generated to perpetuate signaling-based attacks in computer networks. The problem is studied for the Session Initiation Protocol (SIP) using a dataset of signaling packets exchanged by multiple end-users. A sequence of SIP messages never observed before can indicate possible exploitation of a vulnerability and its detection or prediction is of high importance to avoid security attacks due to unknown abnormal SIP dialogs. The paper starts to briefly characterize the adopted dataset and introduces multiple definitions to detail how the deep learning-based approach is adopted to detect possible attacks. The proposed solution is based on a convolutional neural network capable of exploring the definition of an orthogonal space representing the SIP dialogs. The space is then used to train the neural network model to classify the type of SIP dialog according to a sequence of SIP packets prior observed. The classifier of unknown SIP dialogs relies on the statistical properties of the supervised learning of known SIP dialogs. Experimental results are presented to assess the solution in terms of SIP dialogs prediction, unknown SIP dialogs detection, and computational performance, demonstrating the usefulness of the proposed methodology to rapidly detect signaling-based attacks.

**Keywords:** deep learning; multimedia networks; SIP protocol

## 1. Introduction

In the previous years, we have observed the banalization and consequent massification of multimedia services, with greater expression in the recent months due to the needs created during the COVID-19 pandemic period. The Session Initiation Protocol (SIP) [1] plays an important role on the operationalization of multimedia sessions, as it supports a plethora of communication services, including voice calls and legacy Public Switched Telephone Network (PSTN) systems through Voice over Internet Protocol (VoIP) [2]. In cellular networks, SIP is also crucial to support all IP Multimedia Subsystem (IMS) services' signaling [3,4], including multimedia and non-multimedia services. The SIP protocol allows the establishment of sessions through adequate authentication mechanisms and signaling control flows that are dynamic enough to accommodate several purposes, e.g., session initiating, maintaining, and terminating between two peers.

The security of the SIP protocol is of high importance to the telecommunication operators running cellular and PSTN networks, and in supporting non-commercial VoIP services in general. It is well known that SIP exhibits a significant number of vulnerabilities [5,6] associated to the authentication process, malformed SIP messages, and signaling attacks. In this paper, we focus on the vulnerabilities caused by the combination of different signaling patterns, which can cause denial-of-service, unauthorized access to a call, billing errors, and other types of attacks [5]. The identification of potential malicious SIP signaling sequences received by the SIP servers and peers can minimize or even avoid the consequences of hypothetical attacks, in particular, the prediction and detection of new signaling sequences

never observed before. While the already known potential malicious sequences can be detected in an automated way, the SIP sequences never observed before need to be analyzed by domain experts who can then assess their level of vulnerability. However, the detection of anomalous SIP signaling sequences is challenging due to the high number of different signaling sequences, the order of the messages in the dialog, and the dialogs' variable length.

Considering the requirements previously introduced to detect anomalous SIP dialogs, and, consequently, trustworthy SIP dialogs, in this paper we propose a deep learning scheme to detect SIP signaling sequences. Through the adoption of a deep learning scheme, the proposed solution is capable of inferring the most likely SIP dialog identifier based on the knowledge acquired during a training stage with a lower computation complexity. Regarding the architecture adopted for the deep learning model, we propose a scheme based on Convolutional Neural Networks (CNNs) as opposed to [7], where Recurrent Neural Networks (RNNs) are adopted. Although the RNN models were specifically designed to process sequential data, a CNN model can also tackle the proposed problem by considering each temporal sequence as a pattern of SIP messages that compose the SIP dialog. By using a CNN the computation complexity of the neural network can be reduced when compared to the LSTM. Additionally, the LSTM and CNN neural networks' outputs are used in two different probabilistic classifiers to detect SIP dialogs never seen before. The main contribution of the paper relies on the comparison of the performance of the proposed classifiers, showing their effective capacity to mitigate vulnerabilities originated by untrustworthy SIP dialogs never observed before. Although the deep learning models used in the work are not a novelty per se, their adoption in a SIP network scenario is advantageous due to the increased performance in terms of detection probability and computation time.

The contributions of this work are summarized as follows:

- We propose a deep learning scheme formed by a Convolution Neural Network (CNN model) to infer the most likely SIP dialog identifier for each received SIP signaling pattern.
- To identify possible misdetections on the CNN model, a classification scheme is proposed to distinguish between already trained and unknown dialogs. The classifier uses the maximum value of the CNN output vector as a set of input features.
- The proposed methodology is compared with a Long Short-term Memory (LSTM) recurrent neural network model proposed in [7]. The comparison between the two schemes includes the SIP dialogs detection and prediction performance, the amount of time required to detect a SIP dialog, and the detection of unknown SIP dialogs.
- The performance comparison shows that the CNN and LSTM RNN models achieve identical performance in detecting the most likely SIP dialog identifier. However, the CNN model exhibits slightly lower computational times.
- We propose two classifiers to detect abnormal SIP dialogs based on the outputs of the neural networks. The performance of the classifiers is characterized for the LSTM and CNN models, showing an abnormal detection rate up to 97.8% computed in less than 300 ms.

The rest of the paper is organized as follows. Section 2 presents the literature review related to SIP security vulnerabilities and different approaches to minimize them. Section 3 describes the CNN model and the classifiers. Section 4 presents the experimental dataset and characterizes the performance achieved by the proposed methodology. Finally, Section 5 concludes the paper.

## 2. Related Work

SIP [1] is an application-layer protocol designed to initiate, maintain, and terminate multimedia sessions through the exchange of SIP messages between each user agent. Each SIP message can be either a request or a response. Initially, a SIP message must be sent with a request that can be identified by a specific method. In response to one of those methods,

a response SIP message is sent with a specific code. Every SIP request exchanged between agents initiates a SIP transaction, and multiple SIP transactions exchanged between two peers form a SIP dialog, which represents the peer-to-peer relationship over time. A user agent can identify the different dialogs through the SIP Call-ID, i.e., a unique identifier for every dialog's message.

The vulnerabilities of the SIP protocol have been identified in several works such as [5,6], including works based on real-world data about SIP brute-forcing attempts for attacking SIP endpoints [8]. The consequences of a SIP attack include service interruption, service destruction, or unauthorized access to previously reserved computing resources. The SIP protocol is currently massively used to support multimedia sessions and signaling of 4G and 5G networks. Improvements of SIP/IMS security have been proposed recently in [9], where the performance of different authentication schemes is compared. In [10] an amendment to provide mutual authentication was proposed. Block-chain authentication schemes were also introduced in [11] to protect the SIP registration process.

SIP service interruption can be caused by flooding attacks, and different solutions include threshold-based classifiers that compare the traffic patterns with the prior statistics [12,13]. Besides the threshold-based solutions, the works in [14,15] detect flooding attacks through a recurrent neural network and a hidden markov model, respectively. Bayesian Networks were adopted in [16] for SIP dialogs' classification. Furthermore, to infer the existence of an attack in [14] the model proposed considers only the content of each SIP message, while in [15] different features are collected, e.g., number of SIP requests, CPU, and memory usage. Malformed SIP messages are another way of compromising SIP. Malicious SIP messages are usually detected through intrusion detection systems, identification of deviations from a priori statistics [17], or rule-based systems that define how a SIP message should be formatted [18]. Another class of SIP vulnerability, aka SIP signaling vulnerability, take advantage of defective implementations of the protocol, where protocol implementation issues can be explored by sending SIP messages to allow improper authentication mechanisms [5]. A mitigation approach for this type of vulnerability was proposed in [19], where a rule-based methodology is used according to the contextual information of the SIP traffic. More recently, the work in [20] has proposed a methodology based on the SIP sequences and their timings that are then used to detect deviations that may represent vulnerabilities. Although different SIP signaling vulnerabilities have already been proposed in [19,20], this work is not assuming a fixed probabilistic model of the SIP operation. Contrarily, we propose a methodology capable of learning from past SIP sequences, which is used to detect unknown SIP dialogs that can be further categorized by domain experts. Moreover, the vulnerability of the abnormal dialogs can also be evaluated based on prior trustworthy SIP data. Table 1 summarizes each attack using the SIP protocol and the solutions proposed to prevent them.

**Table 1.** Type of TIP attacks addressed in the literature.

| Work | Type of SIP Attack | Proposed Solution |
| --- | --- | --- |
| [12] | Flooding | Threshold-based approach (comparison of traffic patterns with the statistics of the network in normal operation). |
| [13] | Flooding | Threshold-based approach. |
| [14] | Flooding | Recurrent neural networks. |
| [15] | Flooding | Hidden markov model. |
| [17] | Malformed-SIP message | Statistical classifiers (e.g., Euclidean distance, Bayesian). |
| [18] | Malformed-SIP message | Rule-based approach (SDP parser module that interprets each SIP message body and drops the messages that do not follow the rules defined in RFC 4566 standard). |
| [19] | SIP signaling | Rule-based approach (event graphs to model the protocol activities). |
| [20] | SIP signaling | Probabilistic state transition machine (description of normal and abnormal events in each dialog and transaction). |
| [7] | SIP signalling | Recurrent neural network. |

### 3. Materials and Methods

The system model assumed in this work is depicted in Figure 1. We consider a SIP network, where a SIP Source node establishes a signaling session with a Destination node. The source and destination nodes are the peer nodes, also called SIP user agents. The server nodes are responsible for routing the SIP messages between the SIP peers, constituting the so-called SIP path.

The methodology to detect abnormal SIP dialogs can be implemented locally at each SIP server or SIP peer. Additionally, the SIP peers or servers can also send the SIP messages to cloud services running the proposed methodology. The methodology is fed by the information collected in the consecutive SIP messages exchanged between the SIP Source and SIP Destination nodes, represented in the figure by the "SIP gathering" block. Specifically, the "SIP gathering" block is responsible to organize the multiple SIP messages into a SIP dataset that is used for training purposes in an offline manner. The block model "Model training" implements the training of the neural networks and the computation of the abnormal classifiers' thresholds. Additionally, the "SIP gathering" block also feeds the SIP messages processed in the SIP path to the "Detection/Prediction model" block. The "Detection/Prediction model" block implements the neural network models trained offline and provides the input to the abnormal SIP dialogs classifiers represented in the figure by the block "Unknown SIP dialog classifier".

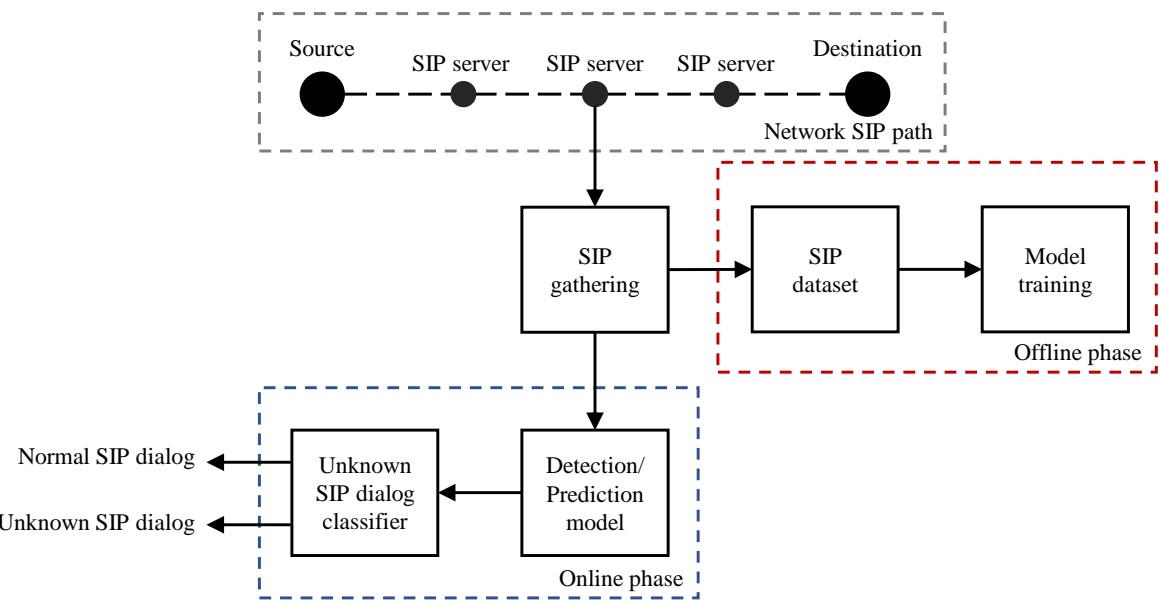

**Figure 1.** System model.

Next we detail the model to perform the detection and prediction of SIP dialogs through their identifiers (problem a), and the identification of new types of SIP dialogs never trained by the model and thus labeled as unknown or abnormal (problem b). The proposed model is able to classify the dialogs already known and label them as safe, anomalous, or according to different vulnerabilities ranks, but also to isolate dialogs never observed before for further analysis, i.e., to posterior send them to a domain expert. To solve the problems mentioned before, the model is divided into two blocks: the "Detection/Prediction model" is described in Section 3.1; the detection of unknown SIP dialogs is presented in Section 3.2.

Figure 2 illustrates the proposed CNN system model. In a nutshell, an observed sequence of SIP messages is initially pre-processed to serve the requisites of the CNN model. After being trained, the neural network of the CNN model provides information to the "Unknown SIP dialog classifier" responsible for detecting if the observed sequence represents a known SIP dialog identifier, or if it is an unknown SIP dialog.

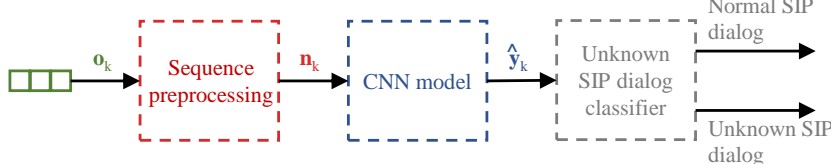

**Figure 2.** Proposed CNN system model.

*3.1. CNN Model*

Before introducing the proposed model, we summarize the notation used in the paper in Table 2. To perform the detection and prediction of a SIP dialog identifier we adopted the CNN model illustrated in Figure 3, comprising the CNN, Max Pooling, Flatten, and Dense layers.

**Table 2.** Table of symbols.

| Symbols | Definitions |
| --- | --- |
| $m_k$ | SIP message $k$. |
| $\mathbf{m}'_k$ | Encoded SIP message $k$. |
| $M$ | Number of all SIP methods and responses. |
| $\mathbf{d}_k$ | SIP dialog $k$. |
| $\mathbf{o}_k$ | Observation $k$. |
| $\mathbf{s}_k$ | Pad sequence of an observation $\mathbf{o}_k$. |
| $L_d$ | Length of a SIP dialog $\mathbf{d}_k$. |
| $L_o$ | Length of an observation $\mathbf{o}_k$. |
| $L_S$ | Length of a pad sequence $\mathbf{s}_k$. |
| $L_M$ | Length of the encoded SIP message $\mathbf{m}'_k$. |
| $n$ | Number of zeros added into the pad sequence. |
| $N$ | Number of unique SIP dialogs. |
| $\mathbf{y}_k$ | Identifier of dialog $k$. |
| $\mathcal{X}$ | Input state space. |
| $\mathcal{Y}$ | Output state space. |
| $Skew(.)$ | Skewness function. |
| $Kurt(.)$ | Kurtosis function. |
| $\lambda_{Max}(k)$ | Maximum average threshold value for SIP dialog $k$ (maximum output-based classifier). |
| $\lambda_S$ | Skewness threshold (skewness and kurtosis-based classifier). |
| $\lambda_K$ | Kurtosis threshold (skewness and kurtosis-based classifier). |
| $H_0$ | Hypothesis 0 (classifier detects a trained dialog). |
| $H_1$ | Hypothesis 1 (classifier detects an unknown dialog). |
| $\mu_S$ | Mean of the skewness of the trained dialogs. |
| $\mu_K$ | Mean of the kurtosis of the trained dialogs. |
| $\sigma_S^2$ | Variance of the skewness of the trained dialogs. |
| $\sigma_K^2$ | Variance of the kurtosis of the trained dialogs. |

Although the CNNs were not originally designed to process temporal sequences as opposed to the Long Short-Term Memory (LSTM) Recurrent Neural Networks (RNNs) adopted in [7], the CNNs can also be trained to recognize a specific SIP dialog.

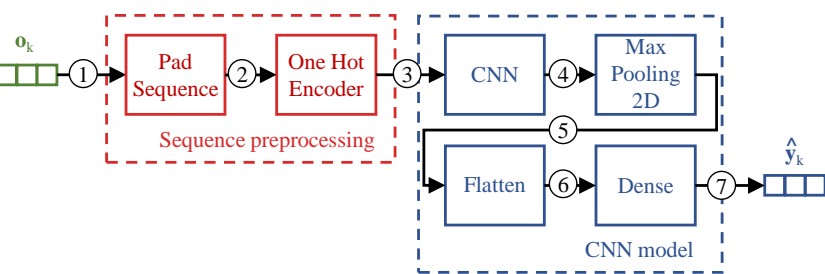

**Figure 3.** CNN model.

Before introducing the model we assume that it can be adopted by the SIP user agents or by the SIP servers traversed by the SIP messages. The first concept to be introduced is the SIP message $m_k$. SIP messages are the atomic unit exchanged by the SIP peers to establish, maintain, update, and terminate a signaling session.

**Definition 1.** *A **SIP message** carried in a SIP packet and denoted by $m_k$, $1 \leq k \leq M$, is a SIP request or SIP response of a specific type. The variable M denotes the total number of unique SIP requests and responses.*

The SIP protocol relies on Request and Response SIP messages. A SIP message can be formed by either a numerical code representing the type of the SIP response or a text field indicating the type of the SIP request. However, to use the SIP message as an input we need to encode each SIP message, since the CNN model can only process numerical values. The encoding process is performed using the One Hot Encoder algorithm [21] to transform each SIP message $m_k$ into a unique Boolean vector orthogonal to the others.

**Definition 2.** *An **encoded SIP message** $\mathbf{m}'_i$ is represented by a Boolean vector with length $L_M$ that univocally identifies the type of the SIP message $m_i$, i.e., $\mathbf{m}'_i = < \underbrace{0, \ldots, 0}_{(L_M - i)}, 1, \underbrace{0, \ldots, 0}_{(i-1)} >$.*

All interactions of a SIP session are implemented through SIP messages, which create different transactions. A SIP session for a specific purpose (e.g., a voice call) is composed of multiple SIP messages. The set of all messages exchanged during the session forms the SIP dialog. A SIP dialog is completed when the multimedia session formed by the multiple transactions is terminated.

**Definition 3.** *A **SIP dialog** denoted by $\mathbf{d}_k = < \mathbf{m}'^{(1)}, \mathbf{m}'^{(2)}, \ldots, \mathbf{m}'^{(L_d)} >$ is formed by a sequence of consecutive SIP messages $\mathbf{m}'^{(j)}$, where j represents the position of the SIP message in the dialog sequence. The length of the SIP dialog is represented by $L_d$. The SIP messages forming the SIP dialog contain the same Call ID string as well as the sender and receiver addresses in the packet's header.*

Although a SIP dialog is only defined when all SIP messages are exchanged, we assume that the model can estimate the dialog when only part of the SIP dialogs' messages has been exchanged. Therefore, instead of considering only sequences with length $L_d$, the model can process their subsequences, i.e., $1 \leq L_o \leq L_d$. Thus, depending on the length of the observation, the CNN model is either predicting ($L_o < L_d$) or detecting ($L_o = L_d$) a SIP dialog identifier.

**Definition 4.** *An **observation** k is a sequence of consecutive encoded SIP messages denoted as $\mathbf{o}_k = < \mathbf{m}'^{(1)}, \mathbf{m}'^{(2)}, \ldots, \mathbf{m}'^{(L_o)} >$, where $L_o$ represents the observation length. To describe the consecutive relation of the messages, each encoded SIP message is represented by $\mathbf{m}'^{(j)} = \mathbf{m}'_i$, $1 \leq i \leq M$. The SIP messages in the observation constitute either a sub or complete dialog and, consequently, they share the same SIP Call ID.*

The observations can have different lengths ($1 \leq L_o \leq L_d$), being transformed into a fixed-length stuffed sequence $\mathbf{s}_k$ defined as follows.

**Definition 5.** *A **pad sequence** $\mathbf{s}_k$ is formed for each observation $\mathbf{o}_k$, by adding n zeros at the end of the observation $\mathbf{o}_k$, i.e., $\mathbf{s}_k = < \mathbf{o}_k, \underbrace{0, 0, \ldots, 0}_{(n)} >$. The length of the pad sequences is denoted by*

$L_S$, *where* $L_S = L_o + n$,.

So far, we have considered that a padding symbol is added to each $\mathbf{o}_k$. Besides the encoding of each SIP message, the padding symbols are also encoded according to Definition 2. Consequently, the encoded SIP message has length $L_M = M + 1$, considering all types of SIP messages ($M$) and the zero-padding symbol.

Next, we describe the input and output state spaces, which are created during the training stage of the CNN model. The state spaces are then used in the prediction/detection stages to map the SIP dialogs with their correspondent identifier.

**Definition 6.** *The **input state space** $\mathcal{X}$ of the supervised learning implemented through the CNN is the set of padded sequences $\mathcal{X} = \{\mathbf{s}_1, \ldots, \mathbf{s}_k\}$, with $k = L_M{}^{L_S}$ learnt in the model's training stage.*

**Definition 7.** *The set $\mathcal{Y} = \{\mathbf{y}_1, \mathbf{y}_2, \ldots, \mathbf{y}_N\}$ represents the **output state space** of the neural network, where $N$ denotes the number of unique SIP dialogs in the training dataset and, $\mathbf{y}_k = < \underbrace{0, \ldots, 0}_{(N-k)}, 1, \underbrace{0, \ldots, 0}_{(k-1)} >$ the identifier of SIP dialog $\mathbf{d}_k$.*

In the detection stage, the CNN model can compute the most likely SIP dialog identifier for a given observed sequence, which can be viewed as a regression problem $\hat{\mathbf{y}}_k = f(\mathbf{s}_k, \beta)$. The estimated SIP dialog identifier $\hat{\mathbf{y}}_k$ is obtained through the estimate function $f(.)$ which is defined by interactively computing the weights of the CNN model ($\beta$) during the training period, so that the input and output state spaces are correctly mapped. The steps followed by the CNN model during the prediction/detection of a SIP dialog identifier are identified in Table 3.

**Table 3.** CNN model.

| | |
|---|---|
| Step 1: | An observed sequence $\mathbf{o}_k$ with length $1 \times L_o \times 1$ is requested to be processed by the system model. |
| Step 2: | The Pad Sequence block appends zeros at the end of $\mathbf{o}_k$, creating a stuffed sequence $\mathbf{s}_k$ with length $1 \times N \times 1$. |
| Step 3: | The One Hot Encoder block encodes each element of $\mathbf{s}_k$ (SIP messages and padding symbol) into an orthogonal Boolean vector. The length $\mathbf{s}_k$ is changed into $1 \times N \times L_M \times 1$. At this stage, the observed sequence is ready to be processed by the CNN model. |
| Step 4: | The CNN layer processes the encoded SIP message $\mathbf{m}'_k$ of the padded sequence $\mathbf{s}_k$ and returns a $1 \times N$ sequence of real numbers in $[0, 1]$. |
| Step 5: | The Maximum Pooling block reduces the dimension of the CNN output by locally selecting the maximum value using a $2 \times 2$ filter. |
| Step 6: | The Flatten block converts the 2-dimensional vector into a 1-dimensional vector. |
| Step 7: | The Dense layer receives the outputs from the Flatten block and generates an output vector $\hat{\mathbf{y}}_k$ of length $1 \times N$ of real numbers in $[0, 1]$. |

Knowing that the field of deep learning is focused on different techniques and model architectures that are hard to compare in a formal way (e.g., regularization, latent space representation, efficient loss functions, gradient-based optimization, etc.), we highlight that the deep learning models used in our work are not a novelty per see and the focus of our work is not on innovative learning models but on their use to derive the classifiers proposed in Section 3.2. In this way, the proposed CNN topology is compared against an LSTM topology proposed in [7]. Tables 4 and 5 describe the parameters adopted in the CNN and LSTM models, respectively. We highlight that more complex models were also evaluated in our work; however, we did not consider them as a solution because they achieved similar classification performance results but have higher computation times.

**Table 4.** Description of the CNN model.

| Layer | Type | Ouput | Activation | Parameters |
|---|---|---|---|---|
| 1 | Conv2D | $56 \times 1 \times 16$ | relu | 592 |
| 2 | MaxPooling | $28 \times 1 \times 16$ | - | 0 |
| 3 | Flatten | 448 | - | 0 |
| 4 | Dense | 928 | softmax | 416672 |

**Table 5.** Description of the LSTM RNN model.

| Layer | Type | Ouput | Activation | Parameters |
|---|---|---|---|---|
| 1 | LSTM | 928 | tanh | 3515264 |
| 2 | Dense | 928 | softmax | 862112 |

*3.2. Unknown SIP Dialogs' Classifiers*

Although the CNN model identifies the most likely SIP dialog identifier given an observed sequence of SIP messages, $\mathbf{o}_k$, we cannot assume that the SIP dialog identifier is always correctly detected. An example of a misdetection can be observed whenever the CNN model detects/predicts the identifier of an unknown observed sequence, i.e., a sequence not considered in the input and output state spaces during the CNN training stage. Thus, two classifiers were developed to distinguish between unknown and trained/known SIP dialogs. The classification is performed considering the output vector from the CNN model $\hat{\mathbf{y}}_k$.

In the first classifier, proposed in [7], the detection of unknown SIP dialogs is based on statistical information computed from the output vectors $\hat{\mathbf{y}}_k$ of the neural network, particularly the skewness and kurtosis standardized moments. Therefore, the first step to be taken is to compute the skewness and kurtosis standardized moments, i.e., $Skew(\hat{\mathbf{y}}_k)$ and $Kur(\hat{\mathbf{y}}_k)$. Then, in the second step the classifier evaluates the statistical information collected from $\hat{\mathbf{y}}_k$ and classifies the SIP dialog as already trained/known dialog (hypothesis $H_0$) or unknown dialog (hypothesis $H_1$). The abnormal SIP dialogs are included in the unknown dialogs, and only a further analysis of all unknown SIP dialogs allows the evaluation of its vulnerability level.

The statistical information of the trained dialogs is used to classify the SIP dialogs observed in real-time. The conceptual model of the first classifier and the steps needed to classify each dialog are illustrated in Figure 4. The classifier evaluates if the skewness and kurtosis of neural network outputs of the trained dialogs are above certain threshold values, given by $\lambda_S = \mu_S - \sigma_S^2$ and $\lambda_K = \mu_K - \sigma_K^2$, respectively. The variables $\mu_S$, $\mu_K$, $\sigma_S^2$, and $\sigma_K^2$ denote the average $\mu$ and variance $\sigma^2$ of the skewness or kurtosis central standardized of the trained dialogs. The hypotheses tested in the second step are written as

$$H_0 : Skew(\hat{\mathbf{y}}_k) \geq \lambda_S, Kurt(\hat{\mathbf{y}}_k) \geq \lambda_K,$$
$$H_1 : Skew(\hat{\mathbf{y}}_k) < \lambda_S, Kurt(\hat{\mathbf{y}}_k) < \lambda_K,$$

where $H_0$ represents the case when an observed SIP dialog is classified as normal and $H_1$ represents the condition to classify it as unknown. Note that the classification is computed through the comparison of the skewness and kurtosis computed from the outputs of the trained neural model for the observed SIP dialog, $Skew(\hat{\mathbf{y}}_k)$ and $Kurt(\hat{\mathbf{y}}_k)$ with the statistics of the trained SIP dialogs, $\lambda_S$ and $\lambda_K$.

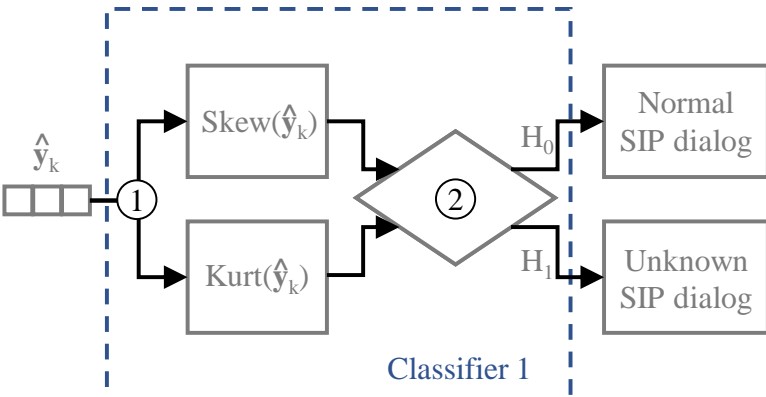

**Figure 4.** Skewness and kurtosis-based classifier.

Rather than computing the standardized moments of the set of features, the second classifier detects possible unknown dialogs by comparing the output inferred by the CNN model $\hat{\mathbf{y}}_k$ with the outputs from the SIP dialogs previously trained. Specifically, whenever an observed SIP dialog is detected, the maximum output-based classifier compares the maximum value of the CNN output vector $\max(\hat{\mathbf{y}}_k)$ with an average threshold value $\lambda_{Max}(k)$ computed with the trained SIP dialogs. The threshold value $\lambda_{Max}(k)$ represents the average of the maximum output value from each SIP dialog labelled as trained dialog. However, the classifier's threshold is not computed based on the average of the maximum output of all trained dialogs. Instead, we compute $N$ average thresholds, once per unique SIP dialog as represented in the vector of thresholds $\boldsymbol{\lambda}_{Max} =< \lambda_{Max}(1), \lambda_{Max}(1), \ldots, \lambda_{Max}(N) = \frac{\sum^{k_N} \max(\hat{\mathbf{y}}_N)}{k_N} >$, where $k_N$ denotes the number of occurrences of dialog $\mathbf{d}_N$ in the training subdataset. Therefore, the classifier compares the maximum value of the predicted SIP dialog identifier $\hat{\mathbf{y}}_k$ with the corresponding threshold value $\lambda_{Max}(k)$. The decision between a trained dialog (hypothesis $H_0$) or an unknown dialog (hypothesis $H_1$) is written as

$$H_0 : \max(\hat{\mathbf{y}}_k) \geq \lambda_{Max}(k),$$
$$H_1 : \max(\hat{\mathbf{y}}_k) < \lambda_{Max}(k).$$

The conceptual model for the maximum output-based classifier is described in Figure 5. Regarding the steps represented in the conceptual model, in the first step, the classifier computes the maximum value of the neural network output. Then, in the second step, the threshold value corresponding to the detected SIP dialog identifier is returned from $\boldsymbol{\lambda}_{Max}$. Finally, with the coefficients assigned, the classifier evaluates which of the hypotheses holds true.

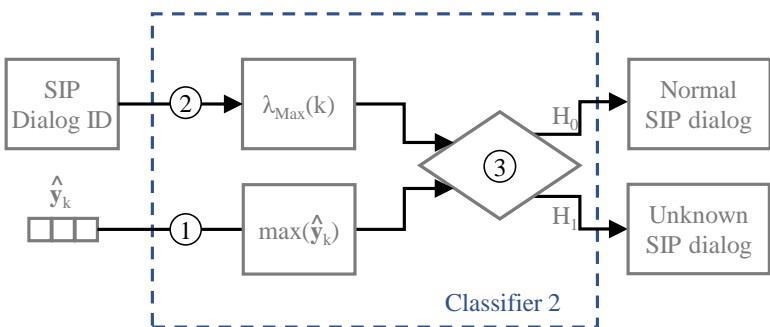

**Figure 5.** Maximum output-based classifier.

## 4. Results and Discussion

To assess the performance of the proposed approach we characterize several performance metrics, including the detection and prediction probabilities, the computation

time, and the classification of unknown SIP dialogs. The results from the CNN model are compared with a LSTM RNN model proposed in [7]. Henceforth, each result reported was obtained using TensorFlow 2.0 running over a 64bit Ubuntu 18.04.5 LTS with 128 GB of RAM and running over an Intel(R) Core(TM) i7-9800X CPU @ 3.80GHz and GeForce RTX 2080 Ti 11GB.

### 4.1. Sip Dialog Dataset

The experimental results were obtained using the dataset created by Nassar et al. in [22], which was also adopted in [7]. The SIP dataset is composed of 18782 SIP dialogs created by 254 user agents, where the 18782 dialogs correspond to 1492 unique SIP dialogs. Each of the 18782 SIP dialogs is formed by a combination of 17 types of SIP messages, which results in dialogs with a length between 3 and 56. To evaluate the performance of our model in different scenarios the dataset was randomly divided into three different subdatasets: training, validation, and testing. The proportions followed for each dataset were the same defined in [7], where the training set is composed of 50% of the original dataset, the validation 20%, and the testing with 30% of all the transactions.

The dialogs were randomly selected from the datasets for training and testing purposes. Furthermore, considering a sample as an ordered sequence of SIP messages observed by a SIP peer/server (SIP dialog), the sampling methodology adopted in the work does not introduce any bias that could affect the results because the dialogs are independent of each other.

Finally, Table 6 describes the parameter values adopted in the proposed model, where some of them are based on the distribution of the dialogs forming the training subdataset, i.e., $M$, $L_M$, $L_S$, $N$. As a consequence, instead of considering 1492 unique SIP dialogs for the variable $N$, only 928 were used. The different values of $N$ are explained by the number of occurrences of each dialog since 66.23% of the 1492 types of SIP dialogs only occur once. Therefore, with the division of the original dataset, some dialogs are specific to each subdataset.

**Table 6.** CNN parameters.

| Model Parameters | |
| --- | --- |
| $M$ | 17 |
| $L_M$ | 18 |
| $L_S$ | 56 |
| $N$ | 928 |
| CNN number of filters | 16 |
| CNN filter size | $2 \times 56$ |
| Dense layer activation function | Relu |
| Max Pooling filter size | $2 \times 2$ |
| Dense layer units | 928 |
| Dense layer activation function | Softmax |
| Early Stopping condition | Minimum of the validation loss |
| Batch size | 64 |
| Loss Function | Categorical cross entropy |
| Optimizer | Adam (learning rate = 0.001) |

### 4.2. Detection and Prediction Performance

The first result to be evaluated is the CNN detection probability, which measures the probability of correctly detecting the SIP dialog identifier when the observed sequence has length $L_o = L_d$, i.e., the multimedia session created by the user agents is terminated. Therefore, to compute the detection probability we compared for each observed sequence $\mathbf{o}_k$ if the output value of the CNN model $\hat{\mathbf{y}}_k$ was similar to the SIP dialog identifier represented in the dataset $\mathbf{y}_k$. Table 7 expresses the detection probability obtained for each subdataset for the proposed CNN model and the LSTM RNN model. To compute the detection probability the CNN and LSTM RNN models were trained during 30 and 255 epochs, respectively.

**Table 7.** Detection probability for each subdataset.

| Model | Train | Validation | Test | Joint |
|---|---|---|---|---|
| LSTM RNN model [7] | 1.0000 | 0.9280 | 0.9354 | 0.9660 |
| CNN model | 1.0000 | 0.9280 | 0.9354 | 0.9660 |

The results show that the CNN model correctly detects all the dialogs in the training subdataset, but it misdetects some of the dialogs in the test and validation subdatasets, as in the LSTM RNN model. The reason for the lower detection performance in the test and validation subdatasets is related to the value $N$, since in the training subdataset there are only 928 types of SIP dialogs instead of 1492. Therefore, the input and output state spaces, created during the training stage, do not characterize all the SIP dialogs leading to the misdetection of some of the dialogs in the test and validation subdatasets.

Besides the detection probability, in Figure 6 we illustrate the prediction probability expressed as a function of the percentage of the SIP dialog's messages sequentially observed so far, i.e., as more SIP messages of the dialog are observed over time. The rationale of Figure 6 is to enable the evaluation of the CNN model performance during the prediction and detection of each SIP dialog regardless of their length, since through the amount of information received we can evaluate the prediction ($L_o/L_d < 100\%$) and detection (($L_o/L_d = 100\%$) performance. Regarding the computation of each curve in the figure, the CNN model processes each SIP dialog considering their observed subsequences, i.e., $\mathbf{o}_1 = < m^{(1)} >$, $\mathbf{o}_2 = < m^{(1)}, m^{(2)} >, \ldots, \mathbf{o}_{L_d} = < m^{(1)}, m^{(2)}, \ldots, m^{(L_d)} >$. Then, we evaluate if the inferred SIP dialog identifier $\hat{\mathbf{y}}_k$ is similar to the true identifier $\mathbf{y}_k$.

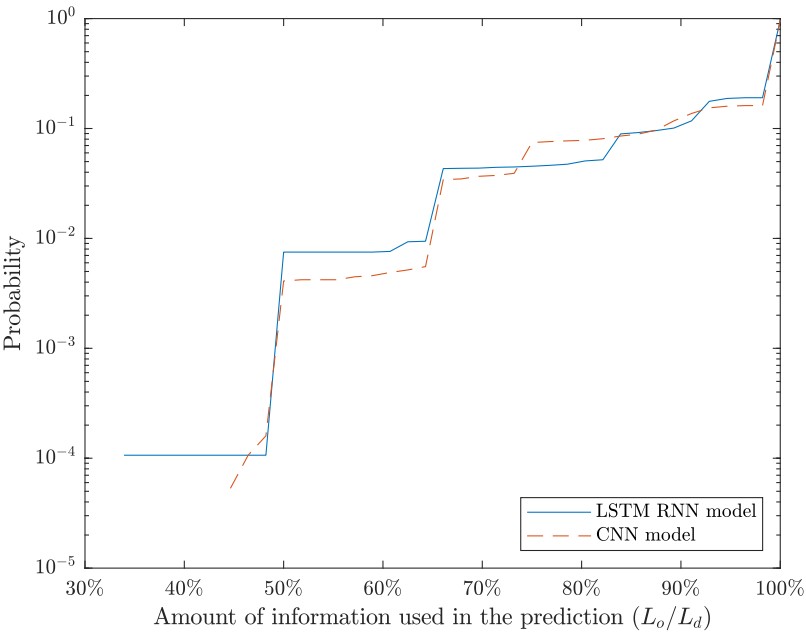

**Figure 6.** SIP dialogs prediction probability over the amount of available information.

The results obtained for the joint datasets (training, test, and validation subdatasets) indicate that the CNN model has a similar performance to the LSTM RNN model. However, the LSTM RNN model is capable of recognizing a higher number of SIP dialogs except when the amount of information received is approximately between 73% and 83%. Additionally, the minimum amount of information to correctly predict a SIP dialog is 44.64% and 33.93% for the CNN and the LSTM RNN model, respectively. Finally, when all SIP messages are received both models achieve a probability of correctly identifying a SIP dialog identifier of approximately 0.9660, which corresponds to the detection probability of the joint dataset, as described in Table 7.

Despite the performance of each model during the detection and prediction of each SIP dialog identifier, in Figure 7 we depict the computation time needed to detect each SIP dialog identifier from the original dataset. Through the measurement and comparison of the computation time, we are able to evaluate the performance of each model in a practical deployment. The computation time represented in the figure is plotted using the Cumulative Distribution Probability (CDF). The results demonstrate that both models have similar behavior, but the CNN model has a lower computation time. Particularly, the average computation time for the CNN and LSTM RNN model is $0.02567s$ and $0.02703s$, respectively. The different computation times are related to the complexity of each model, which can be measured by the number of parameters in the model. The number of training parameters of the CNN model is 646,916 while the LSTM RNN model has 4,377,376 parameters, thus validating the high complexity of the LSTM model.

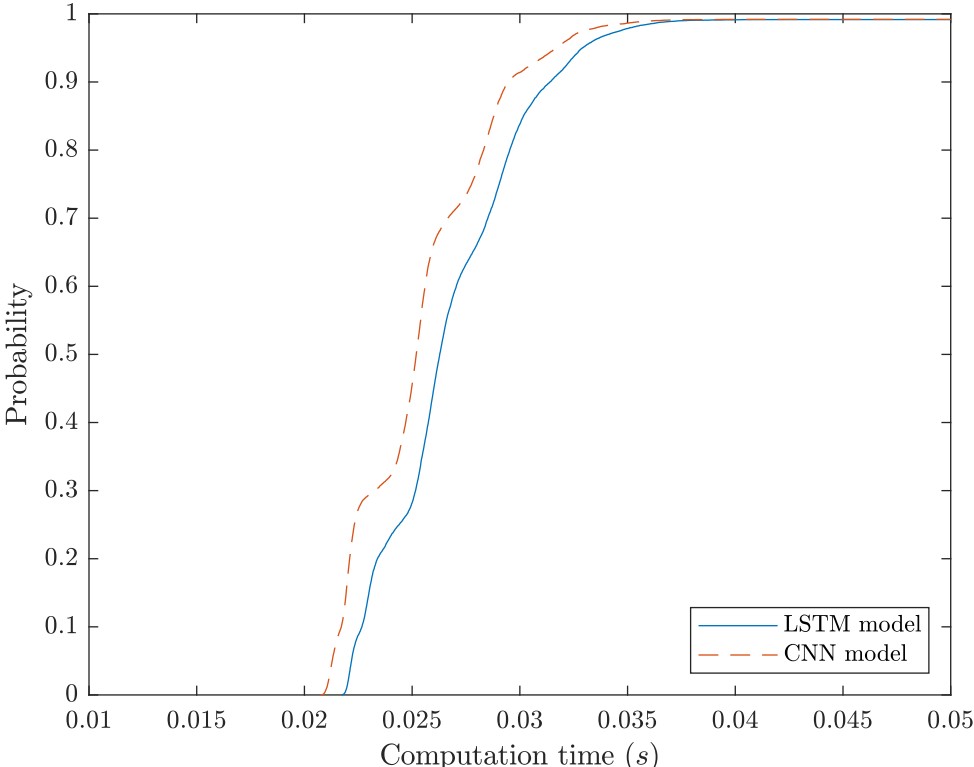

**Figure 7.** CDFs of the detection computation times.

### 4.3. Detection of Unknown SIP Dialogs

As stated the two classifiers presented in Section 3.2 take as input the outputs of the detection model (CNN or LSTM RNN model) and perform the classification based on the SIP dialogs already trained. The outcome of each classifier is a binary output indicating if the observed SIP dialog is a trained/known dialog or a unknown dialog.

The skewness and kurtosis-based classifier uses the statistical information collected from the output of the detection models, which is represented in Figure 8a,b, for CNN and LSTM RNN models, respectively.

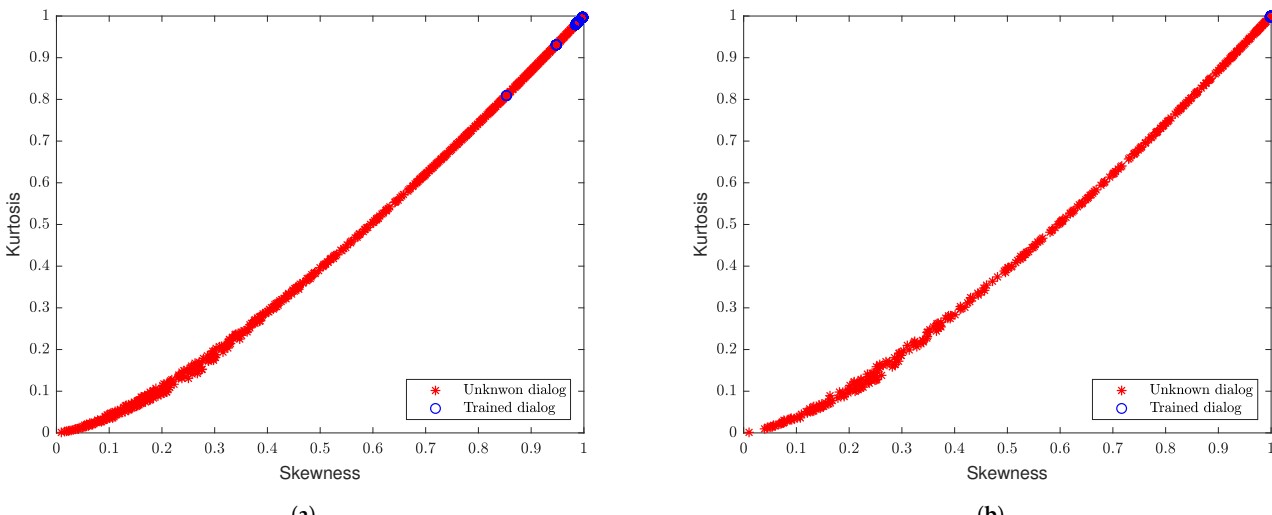

(**a**)                       (**b**)

**Figure 8.** Normalized skewness and kurtosis of the: (**a**) CNN output values (**b**) LSTM RNN output values.

In Figure 8 the features are distinguished by its corresponding label: trained dialog, and unknown dialog. While in Figure 9, we also plotted the threshold values for the skewness ($\lambda_S$) and kurtosis ($\lambda_K$) standardized moments of the CNN and LSTM RNN models given by $\lambda_S = 0.993982$ and $\lambda_K = 0.992029$, and $\lambda_S = 0.999995$ and $\lambda_K = 0.999994$, respectively. According to the threshold values and its plot, we conclude that the trained dialogs lead to lower uncertainty of the neural network outputs and higher values in comparison with the unknown dialogs.

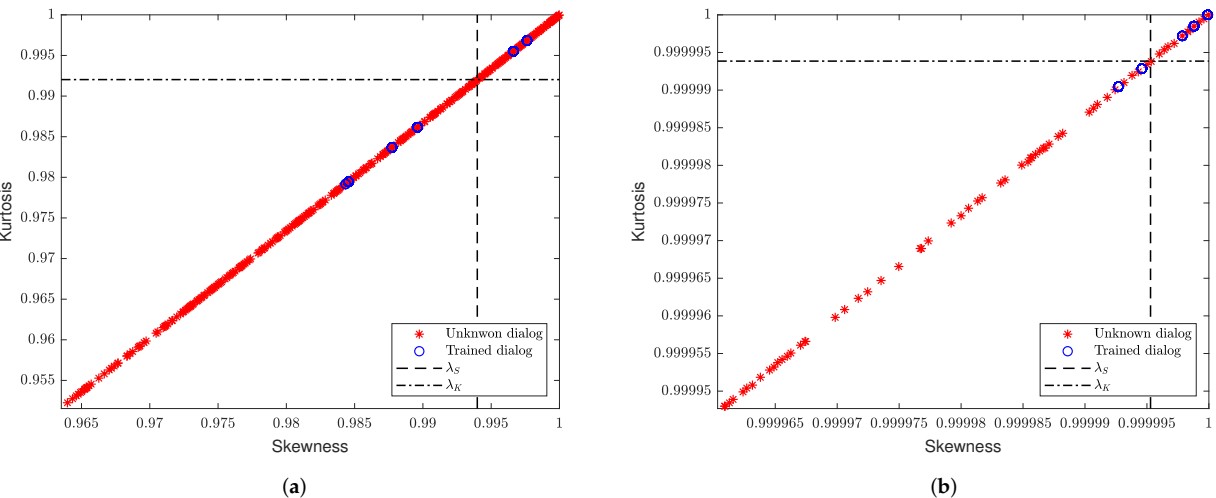

(**a**)                       (**b**)

**Figure 9.** Skewness and kurtosis-based classifier inputs from the: (**a**) CNN model (**b**) LSTM RNN model.

However, for some of the unknown dialogs, the outputs of the neural networks are above the threshold value. The reason for achieving a higher skewness and kurtosis value for some unknown dialogs is related to its similarity to the dialogs represented in each state space, i.e., $\mathcal{X}$ and $\mathcal{Y}$.

To evaluate the performance of the skewness and kurtosis-based classifier, the confusion matrix and other evaluation metrics are represented in Table 8.

**Table 8.** Performance evaluation of skewness and kurtosis-based classifier.

| Model | CNN Model | LSTM RNN Model [7] |
|---|---|---|
| True negative | 0.958333 | 0.964063 |
| True positive | 0.791166 | 0.936721 |
| False negative | 0.208834 | 0.063279 |
| False positive | 0.041667 | 0.035938 |
| Precision | 0.998122 | 0.998648 |
| Accuracy | 0.796933 | 0.937653 |
| F1-score | 0.882675 | 0.966694 |

Four possible outcomes compose the confusion matrix, describing the probability of: correctly classifying a dialog as a trained dialog (true positive), misclassifying a dialog as a trained dialog (false positive), correctly classifying a dialog as an unknown dialog (true negative), and misclassifying a dialog as an unknown dialogs (false negative). The skewness and kurtosis-based classifier achieves a higher performance when used along with the LSTM RNN model, since it reduces the number of unknown dialogs incorrectly classified as a trained dialog. Through the confusion matrix, we computed the remaining metrics: precision, accuracy, and f1-score. However, in none of the remaining metrics, the classification based on the CNN model outputs outperforms the classifier that uses the output from the LSTM RNN model.

In the maximum output-based classifier, the detection of unknown SIP dialogs consists of evaluating if the maximum value of $\hat{\mathbf{y}}_k$ is above the average threshold $\lambda_{Max}(k)$, where the threshold value depends on the SIP dialog identifier $k$ inferred by the detection model. Figure 10 illustrates the Probability Distribution Function (PDF) of the maximum value of the detection model output during the detection of each SIP dialog in the joint dataset. The set of features are identified as trained dialog and unknown dialog.

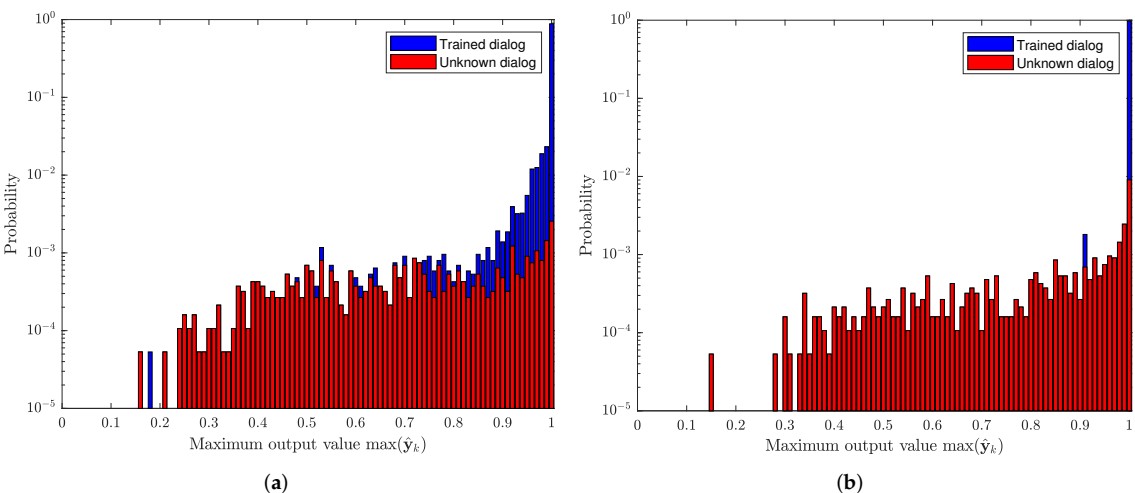

**(a)**

**(b)**

**Figure 10.** PDF of the maximum output considering: (**a**) CNN model (**b**) LSTM RNN model.

Through the distributions presented in Figure 10, we observe that the unknown dialog class tends to have a lower maximum value, while the trained dialog class has a higher maximum value, especially in the LSTM RNN model. Furthermore, comparing the distributions of the trained dialog, we can observe that there is higher uncertainty for the CNN model, while in the LSTM RNN model the majority of these dialogs have a maximum value closer to 1. Together, the conclusions obtained explain why the LSTM RNN model has higher threshold values in the skewness and kurtosis-based classier.

As in the previous classifier, the performance of the maximum output threshold-based classifier is presented in Table 9.

**Table 9.** Performance evaluation of maximum output-based classifier.

| Model | CNN Model | LSTM RNN Model [7] |
|---|---|---|
| True negative | 0.958333 | 0.978125 |
| True positive | 0.639462 | 0.458880 |
| False negative | 0.360538 | 0.541120 |
| False positive | 0.041667 | 0.021875 |
| Precision | 0.997677 | 0.998321 |
| Accuracy | 0.650463 | 0.476573 |
| F1-score | 0.779380 | 0.628753 |

According to the results obtained, the performance of the second classifier exhibits lower performance than the first one. However, there are some similarities between the two classifiers, since the true negative and false positive performance is identical in both classifiers when the features are collected from the CNN model's output. Furthermore, the classification of unknown SIP dialogs (true negative) is improved in the maximum output-based classifier when the features are computed by the LSTM RNN model. To compare the performance of each classifier we used the f1-score metric, which is used when there is a binary classification and the dataset is unbalanced. Therefore, using the f1-score metric, we conclude that the first classifier has a higher performance, achieving an f1-score gain of 10.33% when the set of features is obtained with the CNN model and a gain of 33.94% for the LSTM RNN model.

*4.4. Future Directions*

The computational resources required to run the proposed neural networks are mainly due to the fact that we extend shorter SIP sequences into fixed-length sequences so that fixed-length sequences can be considered in a single neural network model. The proposed assumption increases the neural network complexity because shorter SIP sequences are padded until having the length of the longer SIP sequence already observed and, consequently, the number of inputs and neurons increases due to the use of the padded sequences.

To reduce the computational resources, a different methodology can be explored and based on having multiple neural network models tailored to the length of the SIP sequence, thus avoiding sequence padding. Although the methodology based on multiple neural network models requires higher training times due to the multiple networks associated with the different SIP sequence lengths, it brings two advantages:

- It does not require so complex neural networks, as the number of inputs is significantly lower due to the shorter SIP sequences. Only a single network will have exactly the same number of inputs as the one adopted in the proposed methodology, but it is used to classify a lower number of different SIP sequences.
- Because the different SIP sequences have variable lengths and each network can only tackle a specific length, the number of different sequences to train each network decreases. Consequently, the complexity of each network can be lowered because it classifies a lower number of SIP sequences when compared to a single network used to classify all SIP sequences.

**5. Conclusions**

In this paper, we have proposed a methodology to detect and predict SIP dialogs based on convolution neural networks. The advantage of the CNN model in comparison to the LSTM RNN model proposed in [7] is the lower computational complexity, which leads to a lower computation time during the detection and prediction of SIP dialogs. Several experiments are presented to evaluate the CNN performance against the LSTM RNN model, confirming its advantage in terms of computation time. Furthermore, a classifier based on the maximum output value of the detection model (CNN/LSTM RNN model) was proposed to detect SIP dialogs never observed before. Comparing the performance

of the maximum output-based classifier to the skewness and kurtosis-based classifier, proposed in [7], we conclude that the latter achieved higher performance independently of the detection model. The experimental assessment and the results achieved in terms of detection probability and computation times show the effectiveness of the proposed methodology to improve the security of SIP-based services.

**Author Contributions:** Conceptualization, D.P. and R.O.; methodology, D.P. and R.O.; validation, D.P.; formal analysis, D.P. and R.O.; investigation, D.P.; writing—original draft preparation, D.P. and R.O.; writing—review and editing, D.P. and R.O.; visualization, D.P.; supervision, R.O.; project administration, R.O.; funding acquisition, R.O. All authors have read and agreed to the published version of the manuscript.

**Funding:** This research was funded by Fundação para a Ciência e Tecnologia under the grants UIDB/ 50008/2020, PTDC/EEI-TEL/30433/2017, and PRT/BD/152200/2021.

**Institutional Review Board Statement:** Not applicable.

**Informed Consent Statement:** Not applicable.

**Data Availability Statement:** Publicly available datasets were analyzed in this study. This data can be found here: https://datasets.fbreitinger.de/datasets/ (accessed on 7 January 2021).

**Conflicts of Interest:** The authors declare no conflict of interest.

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
