# Peer review of "Detection of Abnormal SIP Signaling Patterns: A Deep Learning Comparison"

_computers, doi:10.3390/computers11020027_

Round 1
Reviewer 1 Report
The paper presents a comparision between CNN and LSTM-RNN deep learning algorithms to detect abnormal SIP signaling patterns.
The work is interesting, however most of the references on that SIP attacks are really old. The dataset is also quite old and maybe it does not represent the actual situation.
The main concern is that this work highly overlaps with a previous published work, where basic sections are just copied.
Reviewer 2 Report
This paper investigated the detection of abnormal sequences of signaling packets purposely generated to perpetuate signaling-based attacks in computer networks. The problem is studied for the SIP using a dataset of signaling packets exchanged by multiple end-users. The impression of the paper is interesting. However certain points require some illumination in order to fit usual academic standards.
In the abstract, should mention the technique of deep learning that used, CNN model.
In Introduction Section, should be justified why you choose the CNN model in this study, how about other techniques?
In Introduction Section, the contributions points On page 2 should be short, sharp, and describe the contributions of this study better.
In section 2 (Related Work), this section is short, so suggested supporting the section with more studies related to the topic and providing a summary for these studies in Table at the end and show the difference in your study compared with the other studies.
Reviewer 3 Report
I would like to suggest authors consider using some datasets which are more commonly used by researchers like those at https://www.unb.ca/cic/datasets/index.html in addition to the samples created by Nassar et. al in [13]. I then will be happy to review the updates.
Reviewer 4 Report
This study presents a ML methodology for classifying normal vs abnormal/unseen SIP dialogs, the classifier being based on a suitable CNN. This is a step incremental addition to previous work by the authors for the same problem with Bayesian networks and RNNs.
The paper is clearly organized an written, the design of the classifier is explained in depth, and the experimental results validate the assumptions and make clear that the system offers competitive advantage (in terms of computational effort) over the alternative methods. The contribution is interesting and valuable for detection of malfunctioning or malicious SIP agents, so it's worth publishing.
Reviewer 5 Report
In this paper, we have proposed a methodology to detect and predict SIP dialogs based on convolution neural networks. The advantage of the CNN model in comparison to the LSTM RNN model proposed is the lower computational complexity, which leads to a lower computation time during the detection and prediction of SIP dialogs. Several experiments are presented to evaluate the CNN performance against the LSTM RNN model, confirming its advantage in terms of computation time.
There are some issues that the authors should take care of:
Please highlight the main innovations of the paper.
Part of the material and method section has to be explained better paying more attention to the description of the equations so that the paper could be understood by inexperienced readers, too.
I have many concerns about the technical requirements of the testbed environment. I think the model is dependent on very expensive resources and this is a drawback. The authors should point out how can the model reduce the high overhead introduced by their methodology.
Due to the lack of detail in the selection of samples from the dataset, it is unclear whether there is a bias in the samples, selected or accidental, which in turn could affect the results. However, due to the lack of traceability, the strong results cannot unfold and fall behind.
I strongly suggest a figure, that shows the global interaction of each system, as well as a declaration, of what step is upfront (pre-processing), during training and deployment/operation of such system.
The paper has a dense mathematical content. However, the authors do not provide any intuitive or concepts involved behind the mathematical expressions derived. The authors need to provide some insight to make it easier for the reader to understand.
There are many up-to-date theoretical studies on Machine Learning and well-established communities working on different theoretical aspects and techniques (e.g. make your network shallower by fewer layers, use less number of hidden units, decrease regularization, etc). The authors must extend the explanation about the main differences between the current submission and the previous studies. I would suggest a comparison study.
It is really unclear to me the scenario that the authors take into consideration. This is an important point before we can start to reason about the proposed method seriously. We need to understand precisely if the proposed mechanism is able to face the following.
Round 2
Reviewer 2 Report
The authors have addressed the comments.
Reviewer 3 Report
The paper has introduced a few problems of the SIP including Flooding, SIP signaling, Malformed-SIP in Table 1 at Related Work.
However, these problems are not particularly resolved nor a proven solution to improve the defense addressed.
Reviewer 5 Report
The paper is ready for publication
Round 3
Reviewer 3 Report
Clarified.